# Postharvest Quality Improvement of Bell Pepper (*Capsicum annuum* L. cv Nagano) with Forced-Air Precooling and Modified Atmosphere Packaging

**DOI:** 10.3390/foods12213961

**Published:** 2023-10-30

**Authors:** Samuel Yeboah, Sae Jin Hong, Yeri Park, Jeong Hee Choi, Hyang Lan Eum

**Affiliations:** 1Smart Farm Research Center, Korea Institute of Science and Technology (KIST), Gangneung 25451, Republic of Korea; iamsamyeboah@gmail.com; 2Department of Plant Science, College of Life Sciences, Gangneung-Wonju National University, Gangneung 25457, Republic of Korea; dyfl6854@hanmail.net; 3Research Group of Consumer Safety, Korea Food Research Institute, Wanju-gun 55365, Republic of Korea; choijh@kfri.re.kr; 4Postharvest Technology Division, National Institute of Horticultural and Herbal Science, Rural Development Administration, Wanju-gun 55365, Republic of Korea

**Keywords:** bell pepper, fruit color, harvest time, postharvest storage

## Abstract

Optimum postharvest storage conditions increase the postharvest quality and shelf life of horticultural crops. The effects of forced-air precooling (FAP) and modified atmosphere packaging (MAP) on shelf life, physicochemical quality, and health-promoting properties of bell pepper (*Capsicum annuum* L. cv. Nagano) harvested at 90 and 50% coloring stages in May and July respectively, stored at 11 °C, 95% relative humidity were assessed. Fruits were subjected to four treatments: FAP + 30 μm polyethylene liner (FOLO); FAP-only (FOLX); 30 μm polyethylene liner-only (FXLO); and control (FXLX). The quality attributes, viz. weight loss, firmness, color, soluble solids content (SSC), soluble sugars, total phenolic content (TPC), total flavonoid content (TFC), 2,2-dephenyl-1-picrylhydrazyl (DPPH), and 2,2′-azino-bis-3-ethylbenzo-thiazoline-6-sulfonic acid (ABTS) were evaluated. The investigated parameters differed significantly (*p* < 0.05) among treatments except for soluble sugars. FOLO maintained sensory quality (weight loss, firmness, and color), physicochemical (SSC and soluble sugars), and health-promoting properties compared to other treatments during storage. The 50% coloring fruits had a huge variation between treatments than 90% coloring. The results revealed more TPC and antioxidant capacity in the 50% than in the 90% coloring fruits. The study highlights the need to consider the ideal fruit coloring stage at harvest under the effect of FAP and MAP treatments in preserving bell pepper’s postharvest quality and shelf life.

## 1. Introduction

Bell pepper (*Capsicum annuum* L.) is a commercially important fruit vegetable widely grown in the world because of its economic value and consumer demand [1]. However, marketing is constrained by the relatively short shelf life of bell peppers due to water loss, shriveling, and decay during shipment and storage [2,3]. Improving its postharvest quality and shelf life is concomitant with a plethora of benefits, including taste and natural antioxidants, such as phenolic compounds [4,5,6].

Bell pepper, as an emerging export crop, its production is divided into summer and winter cultivation in Korea. Summer cultivation begins in early March and is harvested in June until November, while winter cultivation starts in early September and is harvested in early December until June.

Bell pepper postharvest quality is associated with multiple traits, including visual appearance, flavor, chemical composition, and nutritional value. The maturity of bell peppers solely depends on appearance, firmness, and shelf life, which marketers, exporters, or consumers consider at the spell of the initial purchase [7]. A major constraint in its production is determining the appropriate maturity stage, which is difficult even for fruit with similar physical traits during harvest [5,8]. Immature fruits (onset of turning color) may have unacceptable qualities, while whole-colored fruits lose their attractiveness and firmness, becoming soft in texture within a short time, influencing postharvest quality and consumer preference [9,10,11].

Postharvest losses are a major constraint of bell pepper, estimated as 25–35% of the total production [12]. Temperature has various physiological effects on horticultural crops during shipping, marketing, and storage [12,13]. Metabolic activities such as respiration, transpiration, ethylene production, and shriveling after harvesting of horticultural crops are inclined by temperature, which affects postharvest quality [13,14]. Postharvest deterioration of fruits and vegetables is elevated due to high air temperature and relative humidity, especially during summer cultivation [15]. Therefore, meticulous handling and adequate care are required to maintain postharvest quality.

Forced-air precooling (FAP) is a process of quickly removing field heat from freshly harvested produce by blowing air through the produce [16,17]. Immediate precooling of fruit vegetables after harvest lowers the rate of respiration, minimizes water loss, and slows enzymatic activity and shriveling [18,19], maintaining shelf life and postharvest quality. Antoniali et al. [16] reported that forced-air precooling preserved the postharvest quality (mass loss and firmness) of bell pepper during storage.

Modified atmosphere packaging (MAP) is a technology used to extend the quality and shelf life of horticultural products of high commercial value [7,20,21]. In recent studies using fruits such as pomegranate [22] and persimmon [23], packaging techniques were used to effectively enhance the postharvest quality and shelf life. The packaging of bell peppers in perforated plastic films has been confirmed to reduce water loss, delay softening, and extend shelf life during storage [3,21]. Suitable packaging material such as polyethylene (PE) liner can inhibit biochemical processes and prolong the shelf life of horticultural crops, including pepper, during storage [24,25].

This study assessed the effects of FAP and MAP treatments on the sensory, physicochemical, and health-promoting properties of bell pepper cv. Nagano harvested at 90 and 50% coloring stages during cold storage.

## 2. Materials and Methods

### 2.1. Plant Material and Greenhouse Climate Condition

Bell peppers (*Capsicum annuum* L. cv Nagano) were procured from a greenhouse (37°26′59.5248″ N and 129°9′54.7416″ E), Samcheok, Korea. Fruits were harvested at 90 and 50% based on fruit pericarp (Figure 1A) in May and July, respectively. The fruits were packed in cartons and transferred to Gangneung-Wonju National University Postharvest Biology and Technology Laboratory. Fruit cartons (52 × 37 × 32 cm) weighing about 10 kg were palletized (1 × 2) and precooled at 8 °C for 6 h using (FOX-S1004, DSFOX, Seoul, Republic of Korea) at a wind speed of 3.2–3.6 m/s. Fruits were weighed and subjected to four treatments: Forced-air precooling + 30 μm polyethylene (PE) liner (FOLO); forced-air precooling (FOLX); 30 μm polyethylene (PE) liner (FXLO); and Control (FXLX) were stored at 11 °C and 95% relative humidity for 15 or 16 days for domestic and export market, respectively (Figure 1B).

The air temperature at the Samcheok cultivation area monitored five days before harvesting is shown (Figure 1C,D). The daily average temperature in May, five days before harvesting 90% coloring fruits, was maintained at 11.4–14.6 °C, the maximum temperature was 13.5–17.4 °C, and the internal temperature of the plastic greenhouse on the day of harvesting was 23.3 °C. On the other hand, the 50% coloring fruits’ daily average temperature ranged from 25.6 to 29.4 °C, the maximum temperature was 29.7–33.5 °C, and the internal temperature of the plastic greenhouse increased to 30.3 °C.

### 2.2. Chemicals and Reagents Used for the Study

All chemicals, including solvents used, were of analytical grade. Glucose (99.5%), fructose (contains <0.05 mole % glucose by enzymatic assay) and sucrose (98%), DPPH free radical, ABTS (98%), Folin-Ciocalteu’s phenol reagents, ethyl alcohol, sodium carbonate (99.9%), aluminum nitrate (98%), quercetin 95%, gallic acid (97.5–102.5% titration), potassium acetate (99%), and ascorbic acid were purchased from Sigma-Aldrich (St. Louis, MO, USA).

### 2.3. Quality Assessment of Bell Pepper cv. Nagano

To carry out the experiment, 200 fruits harvested at each coloring stage were used. Six samples were subjected to each treatment for the evaluation of sensory and physicochemical measurements during each storage interval.

Fruit weight loss was measured by the method of Sharma et al. [24]. The weight change was expressed as a percentage of the differences between the initial weight and the weight after the storage period.

Firmness was assessed using a texture analyzer (EZ Test/CE-500N, Shimadzu, Kyoto, Japan) at the equator of the fruit with a 5 mm probe at 120 mm/min crosshead speed and expressed as Newton (N).

The chromaticity of the fruit skin was measured with a colorimeter (CR-400, Minolta, Osaka, Japan) and expressed as Chroma, Hue angle, Hunter *L*, *a*, and *b* values. Hunter *L* value represents lightness (0 = black, 100 = white), Hunter *a* value represents green (-) to red (+), and Hunter *b* value represents blue (-) to yellow (+). Using Hunter *a* and *b* values, the Chroma value was calculated as (C = √ (*a*^2^ + *b*^2^)^1/2^), and the hue angle (H = tan^−1^ (*b*/*a*)). The hue angle of the color wheel represents red-purple, yellow, bluish-green, and blue at 0°, 90°, 180°, and 270° respectively.

The soluble solids content was measured with a refractometer (PAL-1, Atago, Tokyo, Japan) after the flesh tissue was juiced (approximately 0.1 mL) on the lens of the refractometer and expressed as °Brix as reported by Sharma et al. [24].

### 2.4. Extraction Procedure and HPLC Analysis of Soluble Sugars

Finely ground powder (0.2 g) of lyophilized fruit samples was extracted with 12 mL of methanol (50%, *v*/*v*). Samples were vortexed, sonicated at 30 °C for 5 min, and then centrifuged at 3075× *g* at 10 °C for 5 min. An aliquot (1.0 mL) of the supernatants was filtered (0.45 μm syringe filter), followed by an injection of a volume of 5 µL into high-performance liquid chromatography (YL 9100 HPLC, Youngin, Anyang, Republic of Korea), and each elution was performed for 30 min. The separation of sugars was carried out using the Sugar-Pak column (6.5 × 300 mm, 10 μm, Waters, Milford, MA, USA) operated at 30 °C, with 3rd distilled water as a mobile phase, and the flow rate was 0.5 mL/min. A refractive index detector was used to monitor the eluted carbohydrates. The separated sugars were monitored by refractive index and identified by comparing their retention time with standards (fructose, glucose, and sucrose).

### 2.5. Extraction Procedure and Spectrophotometric Analysis of Total Phenolic Content, Total Flavonoid Content, and Antioxidant Activities

Lyophilized samples (0.1 g) were ground into a fine powder and extracted with 1.5 mL of ethanol (70%, *v*/*v*) vortexed and kept in darkness for a day at room temperature (20 °C). It was vortexed again and then centrifuged at 3075× *g* for 5 min at 10 °C and finally filtered (0.45 μm syringe filter) for spectrophotometric (Thermo Fischer Scientific Inc., Waltham, MA, USA) analysis.

Total phenolic content (TPC) was determined according to the Folin-Ciocalteu spectrophotometric method described by Ghasemnezhad et al. [26] and expressed as mg of gallic acid equivalents (GAE)/g dry weight (DW). Total flavonoid content (TFC) was determined by Ghasemnezhad and Ghasemnezhad [27] method and expressed as mg of quercetin equivalent (QE)/g DW.

The DPPH and ABTS radical scavenging assays were performed to assess the antioxidant activity using a method previously described by Zhuang et al. [28] method with some modifications. Briefly, Lyophilized samples (0.1 g) were extracted with 70% (*v*/*v*) ethanol and kept in darkness for a day at room temperature (20 °C). An aliquot of 100 µL of 0.15 mM DPPH and ethanol solution was added to make an equal volume of the sample extracts for 30 min in the dark.

Before obtaining ABTS radical cation (ABTS^+^), ABTS was dissolved in a mixture of 7.4 mM distilled water (DW) and 2.6 mM potassium persulfate in a ratio of 1:1. This solution was then diluted in phosphate-buffered saline to an absorbance of 0.7 (±0.03) at 734 nm and incubated overnight at room temperature (20 °C); 10 µL each of 100, 500, and 1000 ppm of the sample extracts were added to ABTS solution (190 µL) and then kept for 10 min in darkness at room temperature. Spectrophotometry (Thermo Fischer Scientific Inc., Waltham, MA, USA), absorbance was measured at 519 nm and 738 nm for DPPH and ABTS, respectively. The standard used was ascorbic acid.

### 2.6. Experimental Design and Statistical Analysis

The experiment was laid out in a factorial (FAP and MAP treatments) completely randomized block design with three replications. Data were subjected to an analysis of variance (ANOVA) in SAS version 9.1.3 statistical software. Means separation was performed using the Duncan Multiple Range Test (DMRT) (*p* < 0.05), and results were expressed as means ± standard deviation. To assess the relationship among metabolites and antioxidants and different treatment groups, sparse partial least squares-discriminant analysis (sPLS-DA) and heatmaps were performed in MetaboAnalyst (version 5.0), GraphPad Prism (version 10.0) and R (version 4.2.1).

## 3. Results

### 3.1. Weight Loss (WL)

Regardless of the treatments, bell pepper cv. Nagano exhibited weight loss significantly (*p* < 0.05) during cold storage. However, weight loss was higher in control (FXLX) compared to the treated fruits (Figure 2). FXLX fruits had lost 2.8% of their initial weight after 15 days of storage when fruits were harvested at the 90% coloring stage (Figure 2A). The lowest WL was recorded in fruits treated with FOLO throughout the storage period compared to other treatments. Similar results were found in the 50% coloring fruits; FOLO weight loss was less than other treatments (Figure 2B). In general, WL recorded in 50% was higher than in the 90% coloring fruits.

### 3.2. Firmness

The treatments significantly (*p* < 0.05) affected the firmness of bell pepper, ranging from 24 to 29 N during storage. A decrease and increase tendency were observed in all treatments. Fruit harvested at the 90% coloring stage observed a continuous decrease in firmness in FXLX samples (Figure 2C). Treated fruits (FOLO, FOLX, and FXLO) significantly enhanced firmness, reaching a peak of 28, 27, and 28 N on the 9th day of storage, respectively. In 50% coloring fruit, the firmness decreased from 29 to 23 N (Figure 2D). Fruits treated with FOLO enhanced firmness throughout the storage time compared to other treatments. The 50% coloring fruits had a huge variation in firmness compared to the 90% coloring fruit.

### 3.3. Fruit Color

Heatmaps were performed to interpret the changes in color parameters (Hunter *L*, *a*, *b*, Chroma, and Hue angle) of bell pepper cv. Nagano (Figure 3). Treated samples (FOLO, FOLX, and FXLO) had higher L than FXLX throughout the storage time. Regarding Hunter *a* value, FOLO-treated samples recorded the lower value delaying ripening, and FXLX samples quantified the highest *a* value, making them more ripped in 90% coloring fruits. On the contrary, FOLO and FOLX rapidly delayed ripening than FXLO and FXLX after 12 days of storage in 50% coloring fruits. FXLO- and FOLO-treated samples had higher Hunter *b* values than FOLX and FXLX in 90 and 50% coloring fruits, respectively. Since the Chroma parameter is derived from Hunter *a* and *b* values, its patterns were similar to Hunter *a* and *b*. FOLO- and FOLX-treated samples had lower Chroma values than FXLO and FXLX throughout the storage time in 90% coloring fruits. On the contrary, FOLO and FXLO maintained higher values than FOLX and FXLX, making them brighter in 50% coloring fruits. Fruit treated with FOLO maintained a higher hue angle delaying ripening than other treatments until 12 days but showed a gradual decrease with an increase in storage period. FXLX samples quantified the lowest hue angle, resulting in a faster change from green to red color throughout the storage period. A similar trend was observed in 50% coloring fruits, where FXLX samples recorded the least hue angle during storage.

### 3.4. Soluble Solids Content (SSC)

SSC is mainly composed of soluble sugars. The SSC was 6.5 °Brix at the beginning of harvest in 90% coloring fruits. FXLX significantly (*p* < 0.05) increased SSC than treated fruits throughout the storage time (Table 1). The Forced-air precooling and PE liner treatments suppressed the ripening and aging of the fruit. In general, the SSC gradually decreased during the storage period. The FOLO maintained SSC during the storage period compared to other treatments. On the other hand, the initial value of SSC was 6.0 °Brix in 50% coloring fruits. FOLO-treated samples recorded the highest SSC than other treatments.

### 3.5. Soluble Sugar Contents

In the current study, the accumulation of assimilates in the fruit by determining the levels of different carbohydrates (major components: sucrose, glucose, and fructose) were assessed by retention times on HPLC assay. A decreasing trend was observed in sucrose and the reducing sugars (glucose and fructose) in both 90 and 50% coloring fruits with no significant difference (*p* < 0.05) among treatments. However, FOLO-treated samples recorded the highest sucrose content at the end of the storage period in 50% coloring fruits than other treatments (Appendix A).

### 3.6. Bioactive Compounds of Bell Pepper cv. Nagano

The effects of forced-air precooling (FAP) and modified atmosphere packaging (MAP) treatments on the bioactive compounds of bell pepper are represented in Table 1. There were differences in TPC between the treated and the control samples during the early days of storage. FOLO significantly maintained TPC than other treatments at 3 and 12 days of storage. However, TPC slightly decreased as ripening progressed toward the end of the storage period. A similar trend was observed in fruits harvested at the 50% coloring stage. A decreasing trend in TPC was found in fruit harvested at the 90% coloring stage, while an increment in 50% coloring was observed at the end of the storage period. Regarding TFC, there was a significant difference among treatments in fruits harvested at the 90% coloring stage. FOLO-treated samples had the maximum TFC (0.02 mg QE/g DW) than other treatments. TFC increased in FOLO and FOLX, while FXLO and FXLX decreased at the end of storage time. There was no difference in the TFC of fruit harvested at the 50% coloring stage throughout the storage duration (Table 1). Fruit harvested at 90% coloring showed lower values in the accumulation of bioactive compounds than fruit harvested at 50% coloring stage.

### 3.7. Antioxidant Activity

The antioxidant activity during the storage of bell peppers is shown (Table 1). For treatments affecting the DPPH scavenging activity, FOLO-treated fruits gradually showed stronger activity than other treatments at 12 days of storage. However, at the end of the storage time, all treatments maintained equal scavenging activity. A higher DPPH scavenging activity was observed in the 50% coloring samples than in the 90% coloring fruit across all treatments. A significant amount of ABTS^+^ scavenging activity was revealed in FOLO-treated samples than in other treatments. Regardless of the treatments, fruit harvested at 50% coloring presented higher ABTS^+^ scavenging activity than at 90% coloring. The FOLO-treated samples scavenged this radical better than other treatments during the storage period.

## 4. Discussion

Weight loss is one of the paramount factors that adversely affect bell pepper fruit quality during shipment, storage, and marketing [12,29]. In this study, FOLO rapidly reduced weight loss as compared to FXLX samples at the end of the storage period (Figure 2A,B), confirming the efficacy of forced-air precooling and PE liner treatments on bell pepper fruit quality during shipment and storage. This result corroborated with Kabir et al. [30], who observed a reduced weight loss (1.46 wt.%) in bell peppers when precooled and immediately stored in a controlled chamber. The increased weight loss of FXLX might be due to changes in the permeability of cell membranes that made them more sensitive to water loss [31] since the fruits were not precooled and wrapped in PE liners during storage. FOLO-treated samples significantly negated weight loss during the entire storage period, and this is because the water vapor permeability was lower than other treatments, hence preventing more water vapor transfer [32].

The cell wall and cell membrane play an important role in maintaining the quality and visual appeal of detached fruits from the mother plant [12]. The authors highlighted that the fruit surface is a protective barrier preventing water loss and leakage of solutes. Forced-air precooling and PE liner treatments significantly maintained fruit firmness as compared to the control (Figure 2). The gaseous equilibrium between the product and the sealed atmosphere in the treated samples resulted in low oxygen and high carbon dioxide concentrations and thus reduced the activation of the cell wall or tissue softening enzymes, allowing retention of firmness, especially in 50% coloring fruits during storage as confirmed by Chitravathi et al. [33]. On the contrary, the decrease in firmness observed in the control fruits could be influenced by a high respiration rate and weight loss [20]. Consequently, the cell wall softening of fruits is enhanced by enzymes such as pectin methylesterase, which directly influences the level of firmness [32,34]. The gradual increase in liquid exudation observed in FXLO-treated fruit after 8 days of storage (Figure 2B) induced its firmness at 16 days of storage, and this might be due to cell wall autolysis.

Studies have shown that the color change of fresh products packaged under a high CO_2_ and low O_2_ atmosphere was delayed due to reduced ethylene synthesis [21,22,23]. The FOLO treatment was the most effective in preserving the color of the bell pepper and maintaining its visual quality compared to other treatments. Fruit color change is associated with the ripening process and is one of the evaluations of physicochemical development stages. The chlorophyll degradation in 90% coloring was faster and, as a result, increased carotenoid synthesis than in 50% coloring fruits [26].

There was a rapid decrease tendency in SSC of 50% coloring fruit compared to the fruits harvested at 90% coloring. The rate of metabolism in the 50% coloring fruit progressed as the SSC increased. This is because the 90% coloring fruits harvested at low temperature in May developed while the required period for each growth stage elapsed, whereas, in the high temperature in July, the period required for each growth stage was shortened due to the high temperature of the growing season [14]. As a result, carbohydrates, which are photosynthetic products including glucose and starch, could not be efficiently accumulated [35]. The increase in sugar may be due to the breakdown of other complex sugars, such as pectin, which is broken down by either autolysis, microbial enzymes, or both [24,32]. It has been reported that sucrose metabolism-related sugars in horticultural crops mainly include glucose, fructose, and sucrose [4,31,32], which is further explained by Hu et al. and Durán-Soria et al. [36,37]. Sucrose is the most important and primary form of transported sugars in fruits [38]. However, they only account for a smaller proportion of the water-soluble sugars, while the reducing sugars in approximately equal contents reach over 50% [35,39]. After most fruits are separated from the parent plant prior to maturity, various biochemical changes occur during cold storage. Degradation of pectin polysaccharides and starch causes dynamic changes in soluble sugars, mainly the inversion of sucrose into reducing sugars, and starch degradation has a major effect on fruit flavor and nutrition since sucrose, glucose, and fructose are the predominant soluble sugars [36,37]. During fruit ripening, the biochemical changes of carbohydrates are highly coordinated, and a good equilibrium through synergy and antagonism systems is maintained. The reducing sugars in 90% were higher than in 50% coloring fruits. However, the sucrose content was lower, and this is because the low temperature in May, when fruit was harvested at 90% coloring, inhibited the activity of enzymes, and the resultant consumption of soluble sugars as substrate and energy was delayed [2,40]. Inverse sucrose content was observed, and this is because sugar accumulation was induced by a series of complex enzymatic reactions, which depend on maturity and environmental conditions; sugar content varies from one bell pepper sample to another [34]. Therefore, it can be deduced that the less the fruit (50%) color, the more accumulation of sucrose. As maturation progresses, sucrose will be broken down, probably leading to an increase in reducing sugars [36,37].

The increment of bioactive compounds in most vegetables has been ascribed to dehydration, destruction of the tissue cell, and inactivation of enzymes such as polyphenol oxidase, as reported by Ornelas-Paz et al. [41]. However, high CO_2_ concentration can induce abiotic stress, which, in turn, increases phenolic compounds in bell peppers, and the high CO_2_ or low oxygen atmosphere might have resulted in minimal changes in phenolic and flavonoids [27]. In this study, the average outside and inside air temperature of 90% coloring fruits harvested in May, five days before, and on the day of harvest was lower compared to temperatures recorded in July (Figure 1C,D). Higher temperatures recorded in July when the fruits were harvested at 50% coloring might have affected the level of bioactive compounds, and this corresponds with the findings of previous reports on pepper phytochemical changes [4] and the bioactive properties of tomato [17].

The assessment of antioxidant activity by DPPH and ABTS radical assay provides important information about the functional quality of plants. Free radical scavenging is one of the ways through which antioxidants inhibit oxidation, which is the result of the free radicals. In the current study, antioxidant activity was higher, irrespective of the treatments. The antioxidant capacities have been shown to be related to their phytochemical constituents. Hence, the increase in antioxidant activities in 50% coloring could be attributed to the presence of total phenolic content in the fruit [17,40]. A significant strong correlation between TPC and DPPH (r = 0.7266, and r = 0.8761) of fruits harvested at 90 and 50% coloring was observed in the current study respectively, which confirmed Zhuang et al. [28] findings of nine peppers DPPH antioxidant activity that correlated well with their TPC (r = 0.8504). However, Ghasemnezhad et al. [26] reported a higher amount of antioxidant activity in whole-colored bell pepper than in turning color fruit. The study showed the opposite results, where the 50% colored fruit recorded the highest antioxidant activity than 90% colored fruit. The decrease in levels of phenolic compounds at the end of storage in the 90% samples could be due to the rate of maturation. This binds phenols to protein, and the change in the chemical composition of the fruit resulted in low antioxidant activity [17,42,43]. The high antioxidant activity observed in 50% coloring fruits could be due to the strong intensity of solar radiation in the Samcheok cultivation area when the fruit was harvested in July. Such results corroborated with the theories of Cheema et al. and Ornelas-Paz et al. [12,41], who stated that fruit antioxidant properties are associated with temperature due to higher metabolic rate.

Multivariate analysis has been useful in studies regarding bioactive compounds and functional properties in the food industry. To further elucidate our current research, we used spare partial squares-discriminant analysis (sPLS-DA) and heatmaps to assess the segregation among different fruit samples and the relationship among variables. sPLS-DA was conducted to reduce the number of variables to produce robust and easy-to-interpret models. The sPLS-DA score plots showed the treatment’s effect on the studied variables in two harvest stages (90 and 50% coloring) (Figure 4). The sPLS-DA models identified components 1 and 2 in a two-dimensional figure to differentiate between treatment groups according to the parameters, which indicates that each treatment in both harvest times has distinct separation, accounting for 31.9% and 17.9% for fruits harvested at 90% coloring and 25.8% and 47.5% at 50% coloring in May and July respectively. A synchronized three-dimensional figure clearly separates FOLO treatment from other studied treatments in the Appendix A.

Furthermore, the heatmaps constructed to examine the data structure and look for similarities between treatments and the studied variables (Figure 5) show the effect of treatment on the mean abundance of the studied parameters. FOLO-treated fruits at 90 and 50% coloring showed higher mean abundance, especially in bioactive compounds, antioxidant activities, and sugars, while effectively preserving the postharvest quality and shelf life of bell peppers than other treatments.

## 5. Conclusions

Bell pepper is among the important horticultural commodities and needs appropriate postharvest handling due to its high susceptibility to quality deterioration. Therefore, this study investigated the effects of forced-air precooling and modified atmosphere packaging on the quality and nutritional attributes of bell peppers harvested at two coloring stages during cold storage.

The study’s findings indicated that FOLO treatment was highly effective in preserving the quality and nutritional attributes of bell peppers. This treatment substantially delayed the ripening process, reduced weight loss, enhanced firmness, maintained high levels of soluble sugars, and preserved essential bioactive compounds and antioxidant activity. These results emphasized the potential of FOLO treatment as a valuable strategy for enhancing the postharvest quality and nutritional value of bell peppers, contributing to prolonging shelf life and improving quality.

## Figures and Tables

**Figure 1 foods-12-03961-f001:**
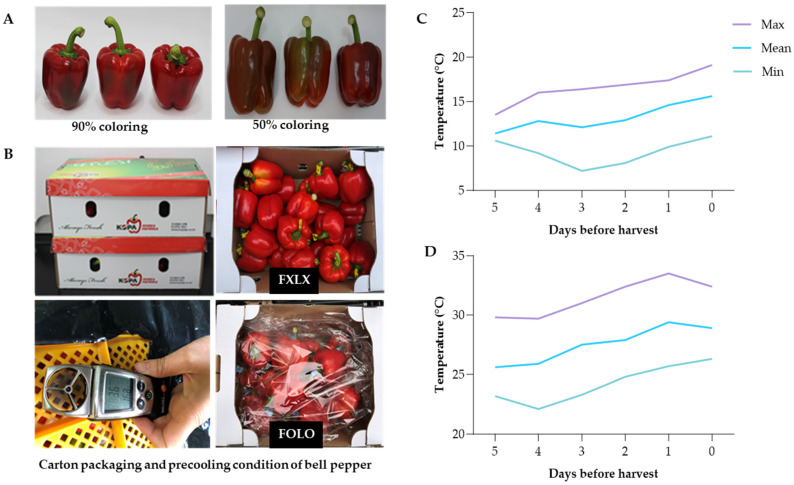
(**A**) The degree of coloring; (**B**) Carton packaging and precooling condition; Temperature five days before harvesting bell pepper cv. Nagano: (**C**) 90% coloring in May and (**D**) 50% coloring in July (**D**) respectively.

**Figure 2 foods-12-03961-f002:**
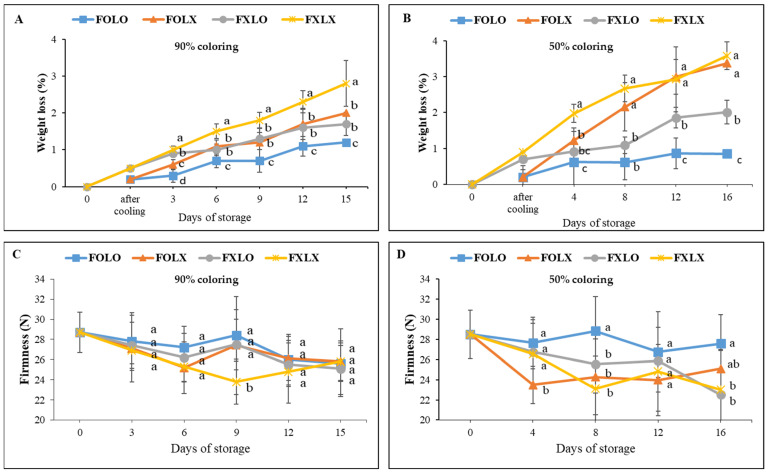
(**A**,**B**) Changes in weight loss; (**C**,**D**) Firmness of bell pepper cv. Nagano during storage at 11 °C, 95% RH. Fruit at 90% coloring (**A**,**C**) and 50% coloring (**B**,**D**) were treated with forced-air precooling or 30 μm PE liner. FOLO, forced-air precooling + 30 μm PE liner; FOLX, forced-air precooling; FXLO, 30 μm PE liner; FXLX, control. Means with different superscripts in the same column are significantly different at *p* < 0.05 (Duncan multiple range test) (*n* = 6).

**Figure 3 foods-12-03961-f003:**
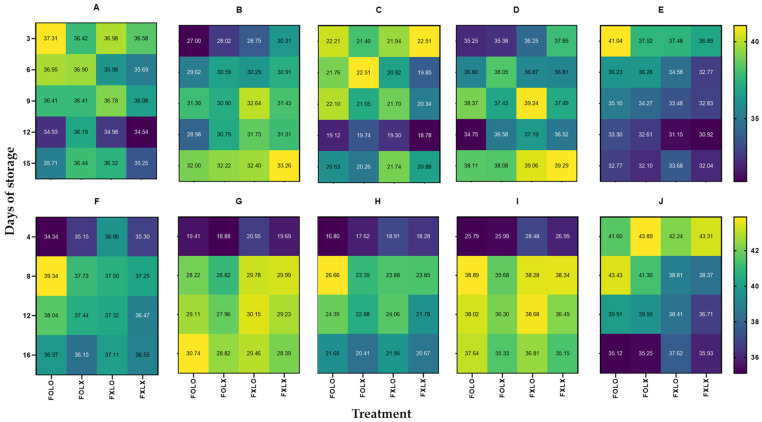
Heatmaps (**A**–**E**) and (**F**–**J**) represent Hunter *L*, *a*, *b*, Chroma, and Hue angle of bell pepper cv. Nagano harvested at 90 and 50% coloring stages and stored at 11 °C and 95% RH for 15 and 16 days, respectively. FOLO, forced-air precooling + 30 μm PE liner; FOLX, forced-air precooling; FXLO, 30 μm PE liner; FXLX, control. The color key indicates an increase or decrease in color change (*n* = 6).

**Figure 4 foods-12-03961-f004:**
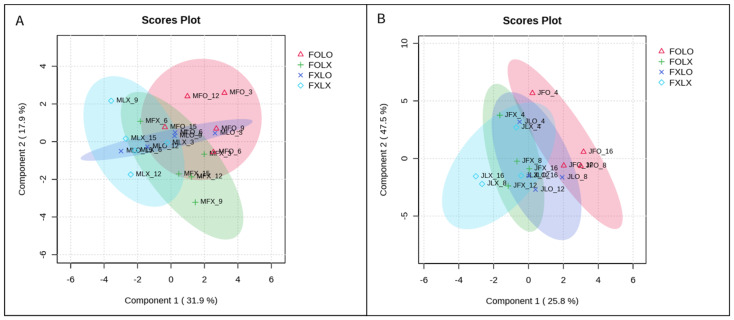
Two−dimensional (2D) sparse partial least squares-discriminant analysis (sPLSDA) scores plot representing treatments of observed parameters of bell pepper cv. Nagano harvested at 90% coloring (**A**) and 50% coloring (**B**) and stored at 11 °C and 95% RH for 15 and 16 days, respectively. FOLO, forced-air precooling + 30 μm PE liner; FOLX, forced-air precooling; FXLO, 30 μm PE liner; FXLX, control.

**Figure 5 foods-12-03961-f005:**
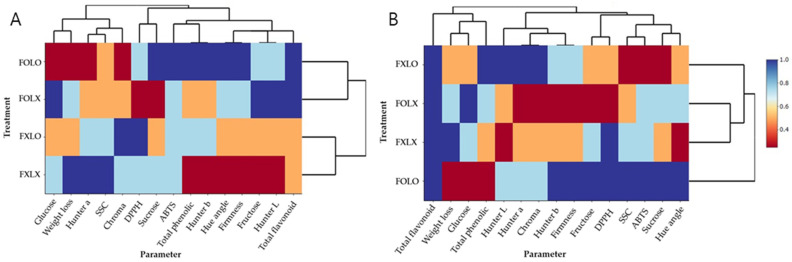
Heatmaps representing treatments and parameters of bell pepper cv. Nagano harvested at 90% coloring (**A**) and 50% coloring (**B**) and stored at 11 °C and 95% RH for 15 and 16 days, respectively. FOLO, forced-air precooling + 30 μm PE liner; FOLX, forced-air precooling; FXLO, 30 μm PE liner; FXLX, control.

**Table 1 foods-12-03961-t001:** Soluble solids content (SSC), total phenolic content (TPC), total flavonoid content (TFC), antioxidant activity by DPPH and ABTS of bell pepper cv. Nagano during storage at 11 °C, 95% RH.

		90% Coloring					50% Coloring			
Parameter	Treatment	3 Days	6 Days	9 Days	12 Days	15 Days	4 Days	8 Days	12 Days	16 Days
SSC (°Brix)	FOLO	6.6 ± 0.3a	6.3 ± 0.2b	6.5 ± 0.4bc	6.3 ± 0.4b	6.3 ± 0.3bc	6.9 ± 0.8a	6.0 ± 0.3b	6.4 ± 0.7a	6.8 ± 0.5a
	FOLX	6.4 ± 0.4ab	6.4 ± 0.3b	6.9 ± 0.6a	6.3 ± 0.1b	6.1 ± 0.7c	6.7 ± 0.7a	6.7 ± 0.6a	5.6 ± 0.3b	6.4 ± 0.2ab
	FXLO	6.1 ± 0.6b	6.3 ± 0.3b	6.3 ± 0.6c	6.9 ± 0.2a	6.6 ± 0.4b	6.1 ± 0.1b	6.5 ± 0.3ab	5.4 ± 0.9b	6.3 ± 0.4b
	FXLX	6.7 ± 0.2a	6.8 ± 0.6a	6.8 ± 0.4ab	6.9 ± 0.6a	7.1 ± 0.7a	6.7 ± 0.5a	6.5 ± 0.3ab	6.4 ± 0.3a	6.2 ± 0.3b
		**	**	**	***	***	*	**	***	**
TPC (mg GAE/g DW)	FOLO	4.74 ± 0.60a	3.97 ± 0.14a	4.31 ± 0.29ab	4.72 ± 0.43a	4.34 ± 0.27a	4.63 ± 0.43a	4.30 ± 0.30b	4.16 ± 0.23b	4.62 ± 0.36a
	FOLX	4.12 ± 0.31bc	4.34 ± 0.21a	3.51 ± 0.13c	4.19 ± 0.64b	4.00 ± 0.40a	4.38 ± 0.06ab	4.50 ± 0.28ab	4.30 ± 0.24b	4.89 ± 0.57a
	FXLO	4.55 ± 0.24ab	4.38 ± 0.68a	4.35 ± 0.23a	4.01 ± 0.29b	4.09 ± 0.31a	4.41 ± 0.14ab	4.95 ± 0.26a	4.68 ± 0.08a	4.40 ± 0.72a
	FXLX	4.01 ± 0.24c	3.98 ± 0.13a	4.09 ± 0.02b	3.74 ± 0.13b	4.21 ± 0.75a	4.17 ± 0.14b	4.72 ± 0.66ab	4.21 ± 0.37b	4.82 ± 0.17a
		**	ns	***	**	ns	*	ns	**	ns
TFC (mg QE/g DW)	FOLO	0.02 ± 0.01a	0.02 ± 0.01a	0.02 ± 0.01 a	0.01 ± 0.00b	0.02 ± 0.00ab	0.02 ± 0.02a	0.01 ± 0.01a	0.02 ± 0.00a	0.03 ± 0.01a
	FOLX	0.01 ± 0.00a	0.01 ± 0.00b	0.02 ± 0.00a	0.03 ± 0.01a	0.03 ± 0.01a	0.02 ± 0.01a	0.02 ± 0.01a	0.01 ± 0.01a	0.02 ± 0.01a
	FXLO	0.02 ± 0.01a	0.01 ± 0.00b	0.01 ± 0.01a	0.02 ± 0.01b	0.01 ± 0.01b	0.02 ± 0.01a	0.02 ± 0.01a	0.01 ± 0.01a	0.02 ± 0.01a
	FXLX	0.02 ± 0.00a	0.01 ± 0.00b	0.02 ± 0.01a	0.02 ± 0.01b	0.01 ± 0.01b	0.01 ± 0.00a	0.02 ± 0.01a	0.02 ± 0.01a	0.02 ± 0.00a
		ns	**	ns	*	**	ns	ns	ns	ns
DPPH Inhibition rate (%)	FOLO	43 ± 4a	42 ± 7b	46 ± 3a	52 ± 7a	48 ± 4a	48 ± 4a	50 ± 6a	47 ± 3a	50 ± 3a
	FOLX	49 ± 5a	49 ± 2a	37 ± 4b	42 ± 6b	44 ± 5a	41 ± 3b	49 ± 3a	50 ± 3a	49 ± 2ab
	FXLO	45 ± 7a	53 ± 5a	49 ± 3a	45 ± 5b	43 ± 5a	47 ± 3a	51 ± 3a	48 ± 3a	46 ± 3b
	FXLX	48 ± 2a	49 ± 3a	45 ± 2a	41 ± 2b	46 ± 7a	47 ± 3a	50 ± 3a	50 ± 3a	50 ± 3a
		ns	**	***	**	ns	**	ns	ns	ns
ABTS Inhibition rate (%)	FOLO	23 ± 2b	25 ± 3ab	26 ± 1ab	27 ± 2a	28 ± 1a	30 ± 1a	27 ± 2a	26 ± 2a	30 ± 1a
	FOLX	25 ± 2ab	27 ± 1a	22 ± 3c	23 ± 1b	22 ± 2b	25 ± 1c	28 ± 3a	27 ± 1a	28 ± 2a
	FXLO	26 ± 1a	20 ± 1c	28 ± 2a	24 ± 1b	22 ± 3b	25 ± 1c	28 ± 1a	26 ± 1a	24 ± 4b
	FXLX	24 ± 2ab	24 ± 1b	24 ± 3bc	23 ± 4b	24 ± 3b	27 ± 1b	28 ± 4a	26 ± 4a	27 ± 1ab
		ns	***	**	*	**	***	ns	ns	**

FOLO, forced-air precooling + 30 μm PE liner; FOLX, forced-air precooling; FXLO, 30 μm PE liner; FXLX, control. Mean values followed by different letter(s) within a column denote the significant difference (*p* < 0.05) by Duncan’s multiple range test (DMRT). ns, *, **, *** not significant or significant at *p* < 0.05, *p* < 0.01, *p* < 0.001, and *p* < 0.0001, respectively. DW (dry weight), SSC (*n* = 6), TPC, TFC, DPPH, and ABTS (*n* = 3).

## Data Availability

The data are available from the corresponding authors.

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
