# Peer review of "Postharvest Quality Improvement of Bell Pepper (Capsicum annuum L. cv Nagano) with Forced-Air Precooling and Modified Atmosphere Packaging"

_foods, 2023, doi:10.3390/foods12213961_

Round 1

Reviewer 1 Report

Comments and Suggestions for Authors

Dear authors,

I have reviewed the paper foods-2682644, entitled "Extension of Shelf Life and Postharvest Quality of Bell pepper (Capsicum annuum L. cv Nagano) with Forced-air Precooling and Modified Atmosphere Packaging". The study is interesting and brings new information to the field.

I have some suggestions of improvement that are presented in the attached pdf file.

Reviewer 2 Report

Comments and Suggestions for Authors

The work is very interesting and innovative, clearly written, easy to follow and understand. I recommended publication with minor revisions:

Maybe the author should describe the preparation ie. packing samples little more clearly.

When you refer to a reference in the text and directly need to mention the name of the author, then it should be e.g in line 340: Although, [23] reported…”, the author should write “Although, Ghasemnezhad et al. [23] reported…”.

The authors should do the same here:

Line 143: “…described by [23]…“; Line 147: “…described by [25] method…” Line 339: “…which confirmed [25] findings…”; Line 144: “…determined by [24] method…”.

The results are clearly presented and the discussion is good. 

Reviewer 3 Report

Comments and Suggestions for Authors

The manuscript entitled "Extension of Shelf Life and Postharvest Quality of Bell pepper (Capsicum annuum L. cv Nagano) with Forced-air Precooling and Modified Atmosphere Packaging" is interesting and well written. The introduction is well written, objective is clear, material and methods are reproducible and the conclusions are supported by data. Thus, in my opinion, only minor details are needed before accepting the manuscript.

1- Line 82, 86.... Please,  the correct form is °C

2- Were the analyzes done in triplicate?

3- Figure 3 is incomplete.

4- Line 391. Please, change "horticultural crops" to "bell peppers"

Reviewer 4 Report

Comments and Suggestions for Authors

Dear author,

Many thanks for your good study. My comments are in the attached file.

Regards

Comments on the Quality of English Language

It is OK. Minor English editing is required.

Round 2

Reviewer 4 Report

Comments and Suggestions for Authors

Dear author,

Many thanks for the revised paper.

Regards

Comments on the Quality of English Language

It is OK.
